# Gender and other potential biases in peer review: cross-sectional analysis of 38 250 external peer review reports

Anna Severin,[1,2] Joao Martins,[3] Rachel Heyard,[4] François Delavy,[2] Anne Jorstad,[4] Matthias Egger ![ORCID] [1,5]

Earlier results from this analysis were presented at the 5th International Congress on Peer Review and Scientific Publication, Chicago, Illinois, USA; September 10–12, 2017.

¹Institute of Social & Preventive Medicine, University of Bern, Bern, Switzerland
²Strategy Support, Swiss National Science Foundation, Bern, Switzerland
³ERCEA A.1, European Research Council, Brussels, Belgium
⁴Data Team, Swiss National Science Foundation, Bern, Switzerland
⁵Research Council, Swiss National Science Foundation, Bern, Switzerland

**Correspondence to**
Dr Matthias Egger;
matthias.egger@ispm.unibe.ch

## ABSTRACT

**Objectives** To examine whether the gender of applicants and peer reviewers and other factors influence peer review of grant proposals submitted to a national funding agency.

**Setting** Swiss National Science Foundation (SNSF).

**Design** Cross-sectional analysis of peer review reports submitted from 2009 to 2016 using linear mixed effects regression models adjusted for research topic, applicant's age, nationality, affiliation and calendar period.

**Participants** External peer reviewers.

**Primary outcome measure** Overall score on a scale from 1 (worst) to 6 (best).

**Results** Analyses included 38 250 reports on 12 294 grant applications from medicine, architecture, biology, chemistry, economics, engineering, geology, history, linguistics, mathematics, physics, psychology and sociology submitted by 26 829 unique peer reviewers. In univariable analysis, male applicants received more favourable evaluation scores than female applicants (+0.18 points; 95% CI 0.14 to 0.23), and male reviewers awarded higher scores than female reviewers (+0.11; 95% CI 0.08 to 0.15). Applicant-nominated reviewers awarded higher scores than reviewers nominated by the SNSF (+0.53; 95% CI 0.50 to 0.56), and reviewers from outside of Switzerland more favourable scores than reviewers affiliated with Swiss institutions (+0.53; 95% CI 0.49 to 0.56). In multivariable analysis, differences between male and female applicants were attenuated (+0.08; 95% CI 0.04 to 0.13) whereas results changed little for source of nomination and affiliation of reviewers. The gender difference increased after September 2011, when new evaluation forms were introduced (p=0.033 from test of interaction).

**Conclusions** Peer review of grant applications at SNSF might be prone to biases stemming from different applicant and reviewer characteristics. The SNSF abandoned the nomination of peer reviewers by applicants. The new form introduced in 2011 may inadvertently have given more emphasis to the applicant's track record. We encourage other funders to conduct similar studies, in order to improve the evidence base for rational and fair research funding.

## BACKGROUND

Expert peer review of research proposals is the accepted best practice for determining which projects are allocated funding.[1] The

### Strengths and limitations of this study

► This study was based on a large sample of peer review reports on project proposals from medicine and other disciplines submitted to the national Swiss funding agency.

► It is one of the few studies examining the interaction between gender of main applicant and gender of reviewers and the 'gender matching hypothesis', as well as the influence of other characteristics of applicants.

► This study only examined scores from peer review, but not the determinants of the final funding decision or the level of funding. It is therefore unclear whether the differences in scores analysed in the present study influenced funding decisions.

► This study was carried out by researchers affiliated with the funding agency and not by an independent group of researchers.

legitimacy of funding decisions relies on a funder's ability to minimise bias in grant evaluations that results from factors that are unrelated to the quality of the applications.[2] Empirical studies suggest that the evaluation of proposals is prone to biases that may relate to both applicant and reviewer characteristics.[2 3] Potential discrimination against women is the most frequently investigated bias.[4] A meta-analysis of 21 studies published from 1987 to 2004 found heterogeneous results, with overall a small gender difference in grant awards, with more men receiving grants than women.[5] More recently, analyses of grant applications submitted to the Canadian Institutes of Health Research from 2012 to 2014 showed that female applicants received lower scores[6] and had lower grant success.[7] Similarly, a study of critiques of applications for renewal of National Institutes of Health (NIH) grants found that reviewers assigned significantly worse priority, approach and significance scores to female than male principal investigators.[8] Finally, the success rate of women applying for European

Research Council Starting Grants, Consolidator Grants or Advanced Grants from 2007 to 2016 was consistently lower than the success rate of men.[9]

Other factors than gender can influence peer review. A study of the Australian Research Council found that applicant-nominated reviewers tended to give better ratings than panel-nominated reviewers.[10] Further, an analysis of data from the Austrian Science Fund suggested that international peer reviewers affiliated with research institutions located in countries known for high scientific productivity were generally more stringent than national reviewers.[11]

The Swiss National Science Foundation (SNSF) supports basic research and use-inspired basic research in all disciplines. The main funding scheme of the SNSF is project funding, which provides support to independent researchers who propose research on self-chosen topics.[12] The proposals submitted to the SNSF are peer reviewed by at least two external experts. The foundation allowed grant applicants to suggest reviewers to evaluate submissions via a 'positive list'. The names put forward on the list were then considered as potential reviewers, after a careful check for conflicts of interest (CoI). The SNSF frequently invites reviewers from abroad to review grant applications. Of note, the SNSF introduced new evaluation forms and guidelines for peer reviewers in September 2011, which we describe in the Methods section.

To gain insights into gender bias and other potential biases in peer review, we analysed the database of the SNSF to examine the determinants of overall scores from external peer reviewers in project funding.

## METHODS
### Evaluation of grant applications at the SNSF
The evaluation consists of four steps.[12] The administrative office first checks eligibility and assigns grant applications to two members of the National Research Council (referee and co-referee) based on their field of expertise. Second, eligible proposals are peer reviewed by external experts. External reviewers were identified in several ways: (1) grant applicants suggested experts via the 'positive list', (2) the referee of the National Research Council suggested reviewers, (3) the SNSF administrative offices proposed experts and (4) experts who declined to review suggested other reviewers.[12] For each application, at least two external reviews were required.

The final choice of reviewers was made by the SNSF. Reviewers from the positive list were chosen only if they had the required expertise and there were no CoI. To exclude any CoI, SNSF employees checked whether reviewers had submitted an application for the same call, whether they had published with the applicants in the past 5 years and whether they work at the same institution or in a closely associated unit. Applicants could also submit a 'negative list' of reviewers who, because of a possible CoI, should not be contacted. Providing a positive or a negative list was optional and the lists could include one or several names.

The peer review forms and assessment scale were changed in September 2011 to simplify the review, and to achieve a more equal distribution of scores, with fewer proposals in the top category. Up to September 2011, peer reviewers were asked to score six criteria: (1) current scientific interest and impact of the project; (2) originality of the work; (3) suitability of the methods; (4) work plan, feasibility, cost; (5) experience and past performance of the applicants and (6) specific abilities of the investigators for the proposed project. Reviewers were asked to 'give a rating and provide explanatory comments' for each of the six criteria. In September 2011, new evaluation forms were introduced,[12 13] which asked experts to review proposals according to three criteria: (1) the applicants' scientific track record and expertise; (2) the scientific relevance, originality and topicality of the proposed research and, in the case of use-inspired research, the research's broader impact and (3) the suitability of the methods and feasibility. Furthermore, peer reviewers were asked to declare any CoI, and given the opportunity to submit confidential comments, which would not be seen by the applicants. Up to September 2011, reviewers scored the overall proposal and each criterion on a scale from 1 to 6: poor (score 1), satisfactory, average, good, very good and excellent (score 6). In September 2011, the scale was changed to poor (score 1), average, good, very good, excellent and outstanding (score 6). The two versions of the peer review form are reproduced in online supplementary text S1. The overall score was attributed by the external reviewers and there were no guidelines on how they should weight the criteria. Applications were not blinded and reviewers were therefore aware of applicant's gender and their track records.

In the third step of the evaluation, the two members of the council (referee and co-referee) assessed the peer review reports and considered them when ranking the application relative to other proposals. In the fourth and final step, referee and co-referee presented their assessment at the meeting of the corresponding section of the council. Each application was then voted on and approved or rejected.[12]

### Data and variables
We analysed the overall scores of external peer review reports submitted from 2009 to 2016. The outcome variable of interest was the overall score of a grant application given by external reviewers. Explanatory variables included meta-data on principal applicants and external peer reviewers, including source of reviewer (applicant-nominated vs SNSF-nominated), gender of the applicant and gender of the reviewer (female vs male) and country of affiliation of the reviewer (Switzerland vs other). The mean ratio of female to male reviewers per grant application was 0.2. Eighteen per cent of the grant applications had male-only external reviewers while only 1% had female-only external reviewers.

SNSF-nominated experts included reviewers who were proposed by the referee, the SNSF office or by experts who declined to review. We also considered the research topic of a grant application as defined by the applicant when submitting their application (see online supplementary table S1), type of institutional affiliation (which included Swiss Federal Institutes of Technology and associated institutions, ie, the ETH domain; Cantonal university and other) and age of the applicant. Finally, we introduced a dummy variable to group applications submitted before and after September 2011.

## Statistical analysis

We used a linear mixed effects model to examine the effect of explanatory variables on the overall peer review scores.[14] This model was chosen because the data are clustered and hierarchical.[15] Grant applications received two or more independent reviews, some reviewers had reviewed more than one application and many applicants had submitted more than one grant application over the study period, causing evaluation scores to be clustered at the levels of research projects, reviewers and applicants. We therefore introduced random intercepts for the identifiers of the reviewer, the applicant and the project in the model, thus taking into account the dependence between clustered scores.[16] Given that $y_{ijk}$ is the overall score given by reviewer i to application j submitted by applicant k, the final model is the following:

$$y_{ijk} = X_{ijk}\beta + u_i + v_j + w_k + \epsilon$$

where $X_{ijk}$ is the matrix with the explanatory variables, β is the regression coefficient vector and $u_i$, $v_j$, $w_k$ are the respective vectors of random intercepts and ε is the vector of random errors. We ran crude and adjusted models. The latter were adjusted for gender of the applicant and reviewer, source of reviewers, country of affiliation of the reviewer, the applicant's age (per 10 year increase), affiliation, nationality (Swiss vs other), the field of research (12 categories) and the period of submission of the proposal (before or after the change in peer review forms and scale). To make adjusted and crude estimates comparable, we performed a complete case analysis by deleting peer review reports with missing values for any of the relevant variables. We examined interactions between the gender of the applicant and the gender of the reviewer, and other variables, by including interaction terms in the linear mixed models. We thus examined the 'gender matching hypothesis', which stipulates that female peer reviewers give higher scores to female researchers and that male reviewers do the same for male applicants.[15] We used likelihood ratio tests to assess the strength of the evidence for interactions.

We present crude and adjusted regression coefficients, which reflect differences in peer review scores with their 95% CI. The notebook of the analysis, including summaries of the different statistical models, is available online at www.git.io/fhaJx.

**Table 1** Characteristics of applicants who submitted grant applications to the Swiss National Science Foundation between 2009 and 2016, stratified by gender

| | Male applicants (n=4514 to 78%) | Female applicants (n=1306 to 22%) |
|---|---|---|
| Age (mean (SD)) | 48.24 (8.63) | 46.23 (8.27) |
| Affiliation | | |
| ETH domain | 1195 (26%) | 219 (17%) |
| Other | 481 (11%) | 224 (17%) |
| Universities (reference) | 2838 (63%) | 863 (66%) |
| Nationality | | |
| Other than Swiss | 1896 (42%) | 573 (44%) |
| Swiss | 2618 (58%) | 733 (56%) |
| Field of research | | |
| Medicine | 1029 (23%) | 317 (24%) |
| Architecture | 146 (3%) | 56 (4%) |
| Biology | 611 (14%) | 129 (10%) |
| Chemistry | 378 (8%) | 76 (6%) |
| Economics | 290 (6%) | 84 (6%) |
| Engineering | 527 (12%) | 74 (6%) |
| Geology | 144 (3%) | 24 (2%) |
| History | 209 (5%) | 68 (5%) |
| Linguistics | 203 (5%) | 102 (8%) |
| Mathematics/physics | 491 (11%) | 56 (4%) |
| Psychology | 223 (5%) | 164 (13%) |
| Sociology | 263 (6%) | 156 (12%) |

The characteristics refer to the first submission of a project grant proposal during the study period. Numbers (%) are shown unless otherwise indicated. Analysis based on 5820 unique applicants without missing values.

## Patient and public involvement

This analysis was based on peer review reports submitted to a national research funder. No patients were involved in developing the research question, outcome measures and overall design of the study. Due to the anonymous nature of the data, we were unable to disseminate the results of the research directly to study participants.

## RESULTS

We analysed the summary scores of 38 250 external peer review reports on 12 294 project grant applications across all disciplines that were submitted from 2009 to 2016 by 26 829 unique reviewers from Switzerland and abroad. The average number of reviews per grant application was 3.1, applicants submitted an average of 2.1 grant applications and reviewers reviewed an average of 1.4 applications. The complete case mixed effects regression analyses were based on 37 989 reviews (99.3%).

## Applicant characteristics

The 12 294 proposals were submitted by 5820 applicants: 4514 (77.6%) men and 1306 (22.4%) women (table 1).

**Table 2** Mean of overall score by groups of applicants and peer reviewers

| Group | No. of peer review reports | Mean overall score (SD) |
|---|---|---|
| Female applicants | 7764 | 4.42 (1.25) |
| Male applicants | 30 455 | 4.63 (1.22) |
| Female reviewers | 7591 | 4.44 (1.26) |
| Male reviewers | 30 659 | 4.63 (1.22) |
| Applicant-nominated reviewers | 8755 | 5.12 (1.00) |
| SNSF-nominated reviewers | 29 495 | 4.43 (1.25) |
| International-based reviewers | 29 423 | 4.71 (1.19) |
| National-based reviewers | 8604 | 4.16 (1.28) |

Results based on 38 250 peer review reports.

Most applicants were based at Cantonal universities, were Swiss and the largest number was from medicine. Female applicants were younger than men and more likely to be affiliated with other institutions (eg, universities of applied sciences, the arts or teacher education) than with the Federal ETH domain or the Cantonal universities. Women were also more likely to work in medicine, the social sciences and humanities (psychology, sociology, linguistics) than in Science, Technology, Engineering and Mathematics (STEM) disciplines or biology (table 1).

### Peer review scores across groups of applicants and reviewers

Distributions of overall peer review scores were somewhat skewed, with applications more frequently being awarded high evaluation scores than low scores (see notebook at www.git.io/fhaJx). Male principal applicants received higher evaluation scores than female principal applicants (table 2). Similarly, the analysis of evaluation scores by gender of the reviewer showed that male reviewers tended to award higher scores than female reviewers. Applicant-nominated reviewers awarded higher scores than SNSF-nominated reviewers, and reviewers affiliated with institutions outside Switzerland awarded higher evaluation scores than reviewers affiliated with Swiss institutions.

There were important differences in evaluation scores across research fields. Grant applications in the natural and technical sciences or in linguistics and history received higher evaluation scores than applications from medicine, sociology or psychology (online supplementary figure S1). Gender differences in scores were more pronounced for some research topics (eg, mathematics and physics and engineering, biology and medicine, sociology) than others (eg, geology, history, psychology). Female applicants were under-represented (below 50%) in all research topics (lower panel of online supplementary figure S1).

Applicants aged 60 years or older received the highest evaluation scores, independent of their gender. For the younger age groups, female applicants consistently received lower evaluation scores than male applicants (online supplementary figure S2). Female applicants were under-represented across all age groups, except for the youngest age group, and representation was particularly low in older age groups (lower panel of online supplementary figure S2). Applications submitted by applicants affiliated with the ETH domain received higher evaluation scores than applications from Cantonal universities or from other research institutions. Gender differences in scores were evident for all three affiliations, and women were under-represented for all affiliations (online supplementary figure S3).

Grant applications submitted by Swiss applicants received slightly lower scores than those submitted by applicants with other nationalities, with a similar gap between genders (online supplementary figure S4). Finally, online supplementary figure S5 shows that, as expected, applications submitted before the new forms were introduced received higher scores than applications evaluated later.

### Linear mixed effects models

Table 3 shows crude and adjusted differences in peer review scores by characteristics of applicants, reviewers and research proposals. In the crude model, the difference between male and female applicants was 0.18 points favouring men. More substantial differences of 0.53 points were observed for source of reviewer (0.53 points higher if the reviewer was nominated by the applicants) and country of affiliation of the reviewer (0.53 higher for reviewers from outside Switzerland). Substantial differences were also observed across disciplines. For example, scores were on average 0.68 points higher in mathematics and physics than in medicine, but 0.12 point lower in psychology than in medicine (table 3). Compared with crude differences, most adjusted differences were smaller. For example, the adjusted difference between male and female applicants was reduced from 0.18 to 0.08 points. One exception was the difference observed between proposals evaluated before or after the introduction of the new peer review forms in September 2011 (0.43 points higher scores before the introduction in both analyses).

### Interactions between gender of the applicants and other variables

We examined possible interactions between the genders of the applicants with the other fixed-effect variables in the model shown in table 2. In other words, we examined whether the differences observed between female and male applicants varied across the levels of the other variables. We found that male reviewers gave higher scores both to male and female applicants than female reviewers, but this difference was considerably greater for male than for female applicants. Figure 1 shows the predicted values of the overall score from the bivariable model (p=0.011 from test of interaction). There was some evidence that the gender difference in scores

**Table 3** Crude and adjusted differences in external peer review evaluation scores by characteristics of applicants, reviewers and research proposals

| Variable | Number of reviews analysed | Unadjusted difference (95% CI) | P value | Adjusted difference (95% CI) | P value |
|---|---|---|---|---|---|
| Gender of the applicant | | | <0.001 | | <0.001 |
| Male | 30263 | 0.18 (0.14 to 0.23) | | 0.08 (0.04 to 0.13) | |
| Female | 7716 | 0 | | 0 | |
| Gender of the reviewer | | | <0.001 | | <0.001 |
| Male | 30442 | 0.11 (0.08 to 0.15) | | 0.08 (0.05 to 0.11) | |
| Female | 7537 | 0 | | 0 | |
| Source of nomination of reviewer | | | <0.001 | | <0.001 |
| Applicant | 8688 | 0.53 (0.50 to 0.56) | | 0.49 (0.46 to 0.51) | |
| Office | 29291 | 0 | | 0 | |
| Country of affiliation of reviewer | | | <0.001 | | <0.001 |
| Outside Switzerland | 29384 | 0.53 (0.49 to 0.56) | | 0.47 (0.44 to 0.50) | |
| Switzerland | 8595 | 0 | | 0 | |
| Age of the applicant | 37989 | | <0.001 | | <0.001 |
| Per 10 year increase | | 0.06 (0.03 to 0.08) | | 0.05 (0.03 to 0.07) | |
| Affiliation of the applicant | | | <0.001 | | <0.001 |
| ETH domain | 9960 | 0.30 (0.26 to 0.34) | | 0.11 (0.07 to 0.16) | |
| Other | 4075 | −0.24 (−0.30 to −0.19) | | −0.19 (−0.25 to −0.14) | |
| Universities | 23944 | 0 | | 0 | |
| Nationality of the applicant | | | 0.155 | | 0.143 |
| Other than Swiss | 16545 | 0.03 (−0.01 to 0.06) | | −0.03 (−0.06 to 0.01) | |
| Swiss | 21434 | 0 | | 0 | |
| Field of research | | | <0.001 | | <0.001 |
| Medicine | 7540 | 0 | | 0 | |
| Architecture | 1391 | 0.13 (0.03 to 0.24) | | 0.15 (0.05 to 0.25) | |
| Biology | 3872 | 0.30 (0.24 to 0.36) | | 0.27 (0.21 to 0.33) | |
| Chemistry | 3244 | 0.46 (0.39 to 0.53) | | 0.24 (0.17 to 0.31) | |
| Economics | 2171 | −0.09 (−0.17 to −0.01) | | −0.01 (−0.09 to 0.06) | |
| Engineering | 4880 | 0.32 (0.25 to 0.38) | | 0.07 (0.00 to 0.13) | |
| Geology | 1167 | 0.50 (0.39 to 0.60) | | 0.25 (0.14 to 0.35) | |
| History | 2053 | 0.35 (0.27 to 0.44) | | 0.32 (0.24 to 0.40) | |
| Linguistics | 2244 | 0.30 (0.22 to 0.38) | | 0.26 (0.18 to 0.34) | |
| Mathematics/physics | 3979 | 0.68 (0.62 to 0.75) | | 0.45 (0.39 to 0.52) | |
| Psychology | 2458 | −0.12 (−0.20 to −0.05) | | −0.08 (−0.15 to 0.00) | |
| Sociology | 2980 | −0.06 (−0.13 to 0.02) | | 0.01 (−0.06 to 0.08) | |
| Introduction of reviewer guidelines | | | <0.001 | | <0.001 |
| Before introduction | 11151 | 0.44 (0.41 to 0.47) | | 0.43 (0.40 to 0.46) | |
| After introduction | 26828 | 0 | | 0 | |

Results from linear mixed effects models based on 37979 complete peer review reports.

became larger after the introduction of the new evaluation form (p=0.065, figure 1). There was strong evidence for an interaction (p<0.0001) between gender of the first applicant and his or her affiliation: the gender differences in scores were smallest for applicants based at one of the Cantonal universities, larger for the ETH domain and most pronounced for other institutions of higher education (eg, universities of applied sciences, the arts or teacher education, see figure 1). The interaction p values from the adjusted models were 0.037 (gender of peer reviewer), 0.003 (affiliation of applicant) and 0.033 (change of evaluation form). All p values from the

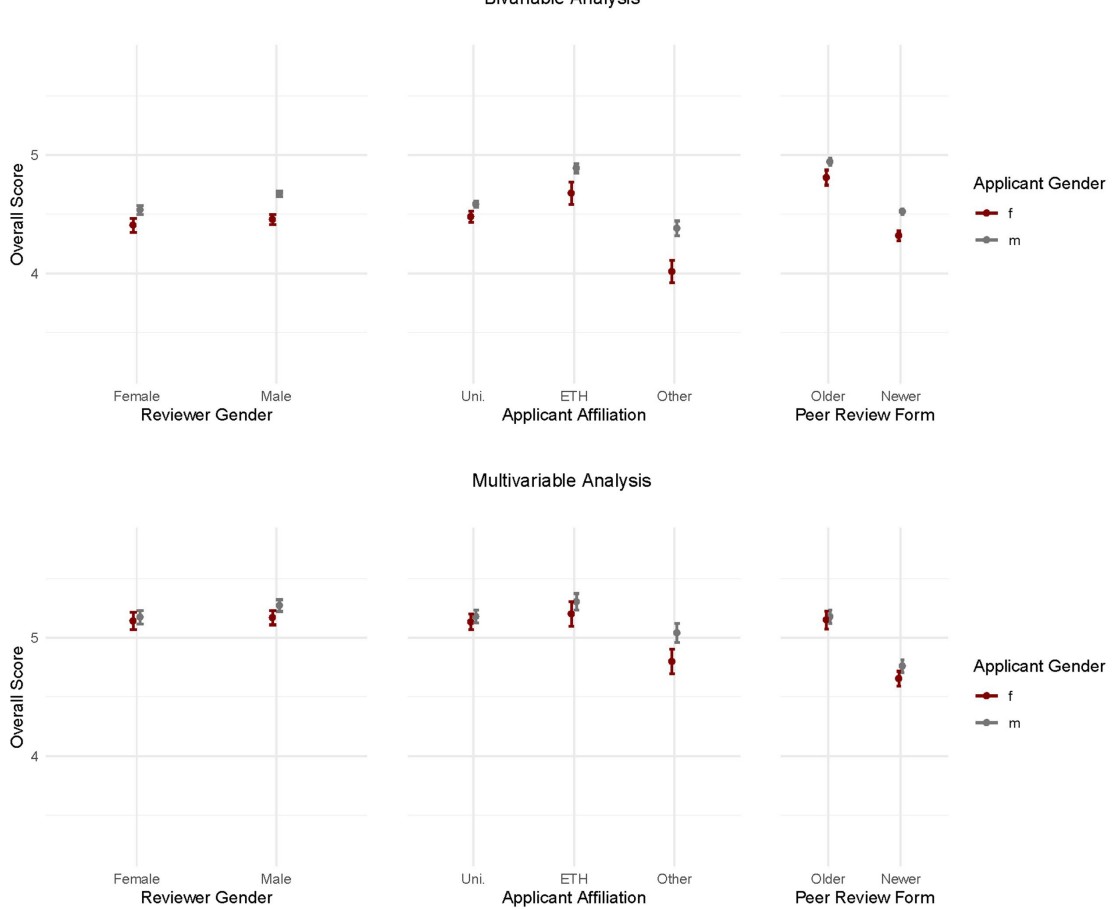

**Figure 1** Gender differences in external evaluation scores by gender of the expert reviewer, affiliation and period of submission of the proposal. Predicted values from bivariable, unadjusted models (upper panel) and the multivariable analysis (lower panel) are shown, together with their 95% CIs. Scores range from 1 (worst) to 6 (best). Average (mean) overall scores are shown, horizontal lines indicate Wald 95% CIs.

bivariable and multivariable interaction tests are shown in online supplementary table S2. Interaction effects were generally small. The effect sizes can be found in the online notebook at www.git.io/fhaJx.

## DISCUSSION

This study of 38 250 distinct grant reviews of 12 294 proposals across all disciplines, which were submitted to the SNSF between 2009 and 2016 by 5832 applicants is to the best of our knowledge one of the largest studies of peer review reports on research proposals ever conducted. Female applicants received lower scores than male applicants. The gender difference was attenuated in multivariable analysis: it was partly explained by the fact that women were under-represented among applicants in the fields and institutions whose proposals were rated highly, for example, mathematics and physics, and institutions of the ETH domain. Our finding is in line with a text analysis of critiques of funded and unfunded NIH grant applications, which found that reviewers assigned significantly worse scores for research approach, significance and priority to female than male applicants. The authors

concluded that reviewers implicitly hold male and female applicants to different standards of evaluation.[8]

Although a substantial proportion of the gender gap in our study was explained by other factors, these factors might be a reflection of the leaky pipeline, that is, 'the phenomenon of women dropping out of research and academic careers at a faster rate than men',[17] which is well documented for Switzerland.[18 19] The academic pipeline in Switzerland is particularly leaky in the life sciences, social sciences and humanities. In STEM the rate of dropout of women is less pronounced, but they are a minority from the start: among PhD students only about 20% are women, whereas in the social sciences, humanities and the life sciences the majority of doctoral students are women.[19]

A noteworthy finding of our study was the interaction between the gender of applicants and peer reviewers. In contrast to Jayasinghe and colleagues,[15] who analysed 7153 reviewer ratings at the Australian Research Council large grant programme and other smaller studies,[2 20] we found evidence supporting the 'gender matching hypothesis'. Male reviewers gave systematically higher ratings to male applicants than to female applicants, whereas the

same phenomenon could not be observed for female reviewers. If such matching bias was present, male reviewers will have favoured male applicants, despite the fact that the proposals from male and female applicants were of similar quality. Alternatively, assuming proposals from male applicants were in fact stronger, female reviewers could have been biased against men and could have downgraded their proposals.

Male reviewers may have given more weight to the track record of applicants than female reviewers. In this context, it is interesting that the gender gap became wider after September 2011, when new evaluation forms for external peer review were introduced. The new guidelines and form separated the criteria related to the applicants, and the criteria related to the proposed project. On the new form, the applicant's track record was the first criterion out of a total of three, whereas it was the fifth out of six criteria on the old form. Although this was not intended, the reform may have led to reviewers giving more weight to the track record of applicants, due to its prominence on the new form. Commenting on a Canadian study, which showed that the gender gap in grant funding was due to less positive assessments of women as principal investigators whereas the quality of the proposed research was similar for women and men,[21] Raymond and Goodman asked funders to 'evaluate projects, not people'.[22] We are planning additional analyses to examine whether at the SNSF the same phenomenon is at play, that is, whether the gender gap is driven by the assessments of the track record. Furthermore, the SNSF is discussing changes to the peer review form.

Our results confirm those from the Australian Research Council, which showed that applicant-nominated reviewers gave higher ratings than panel-nominated reviewers.[10] A study of peer review in biomedical journals also found that author-nominated reviewers submitted more favourable recommendations than editor-nominated reviewers.[23] This difference may be interpreted in several ways. First, nominated reviewers may have a CoI that remained undetected in the SNSF CoI examination. Alternatively, applicants may nominate reviewers who are more familiar with their field than reviewers nominated by the SNSF, and thus more able to recognise the impact and importance of the proposed research. Like the Australian Research Council, the SNSF felt that bias was the more likely explanation and decided to discontinue the use of the 'positive list' in 2016. Of note, applicants can still submit a 'negative list' of reviewers that should not be used because of perceived CoI.

The gender effect was larger for proposals affiliated with an institution from the Federal ETH domain, and especially, from other institutions (eg, universities of applied sciences, the arts or teacher education) compared with applicants affiliated to Cantonal universities. In this context, male applicants from other institutions got systematically higher ratings than their female peers, while the observed gender differences in scores for applicants from Cantonal universities were less pronounced, especially after adjustment for confounding variables. The under-representation of female researchers in the ETH domain and in other institutions might have contributed to this situation, by making the few women applicants appear less qualified to the male reviewers.

Peer reviewers affiliated with a Swiss research institution gave lower scores than reviewers from outside Switzerland. A study of the Austrian Science Fund suggested that reviewers from countries with high scientific productivity were more stringent than national reviewers.[11] Switzerland belongs to the most productive countries in terms of research output[24] and this might explain why reviewers affiliated with Swiss research institutions award lower evaluation scores than reviewers from abroad. In contrast to the Austrian study,[11] the Australian data showed that reviewers affiliated with an institution in the USA were more lenient than reviewers affiliated with institutions located in the UK, Germany or Australia,[25] despite the fact that the USA is the country with the highest research output globally.[24] Other explanations for the lower scores awarded by Swiss reviewers include greater knowledge of the local research capacity and expertise, or bias, if reviewers based in Switzerland downgraded the proposals of their competitors.

Our study has several limitations. First, we did not examine the determinants of the final funding decision or the level of funding. It is therefore unclear whether the differences in scores analysed in the present study influenced funding decisions. Such analyses are planned for the future. Second, this is an observational study and it is therefore difficult to infer causality from the associations observed. Chance, bias and confounding variables must be considered as possible explanations for associations between reviewer and applicant characteristics and overall scores.[26] We tried to control for confounding by adjusting for these variables in regression models. We are considering randomised experiments to test certain interventions (eg, blinding) in order to prevent or reduce gender effects for the future. Third, our results are relevant to the Swiss context, but may not be applicable to other countries. Fourth, we did not attempt to rate the expertise of the reviewers, and adjust for the differences in individual reviewers scores based on their previous performance. We also did not measure the scientific productivity of applicants, and adjust scores for productivity. Other studies have shown that women have lower productivity than men.[6 27] Fifth, this study was carried out by researchers affiliated with the SNSF and not by an independent research institution. As studies might be influenced by the expectations of the researchers of the study, the credibility of the results might be reduced. We address this by making the data available for replication. Finally, we examined project funding only, but not career funding or programme funding.

## CONCLUSIONS

In conclusion, our results had important implications for the evaluation of project grant proposals at the SNSF. The foundation abandoned the nomination of peer reviewers by applicants, and made members of evaluation panels aware of the other factors, including the gender and affiliation of reviewers, that can influence review scores. We encourage all funding bodies to contribute to research on potential biases in research funding, and ways of preventing them.[28]

**Acknowledgements** We are grateful to Angelika Kalt, Benjamin Rindlisbacher, Barbara Curdy-Korrodi and two expert reviewers for helpful comments on previous versions of this paper, and to Andreas Limacher and Lukas Bütikofer (Clinical Trials Unit of the Faculty of Medicine of the University of Bern) for advice on the statistical analyses.

**Contributors** AS, JM and ME conceived the study. JM and RH performed statistical analyses. FD and AJ contributed to data management and statistical analyses. AS and JM wrote the first draft of the paper, which was revised by ME, AS and RH. All authors contributed to and approved the final version.

**Funding** This work was supported by the SNSF (internal funds and grant number 174281).

**Competing interests** None declared.

**Patient and public involvement** Patients and/or the public were not involved in the design, or conduct, or reporting, or dissemination plans of this research.

**Patient consent for publication** Not required.

**Ethics approval** Under Swiss law, not ethics approval is required for this type of study. Peer reviewers did not provide consent. No peer reviewer, applicant or proposal can be identified from this report.

**Provenance and peer review** Not commissioned; externally peer reviewed.

**Data availability statement** Data are available upon reasonable request. The data analysed in this study are available to others on request for an approved research project, after signing a data sharing agreement.

**ORCID iD**
Matthias Egger http://orcid.org/0000-0001-7462-5132

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
