## [Reviewer comments · BMJ Open]

ARTICLE DETAILS

TITLE (PROVISIONAL)	Gender and Other Potential Biases in Peer Review: Cross-sectional Analysis of 38,250 External Peer Review Reports
AUTHORS	Severin, Anna; Martins, Joao; Heyard, Rachel; Delavy, François; Jorstad, Anne; Egger, Matthias

VERSION 1 – REVIEW

REVIEWER	Dr Robyn Tamblyn McGill University CANADA
REVIEW RETURNED	09-Dec-2019

GENERAL COMMENTS	This manuscript provides interesting policy relevant results about potential biases in peer review of grants. Recommended modifications 1. There are sentinel articles on gender effects of reviewers and applicants that are missing from the reviewed literature.2. Both Wenneras and Tamblyn used scientific productivity measures to adjust for differences in the quality of the applicants. (Wennerås C, Wold A. Nepotism and sexism in peer-review. Nature 1997;387:341-3 , Tamblyn R, Girard N, Qian C, Hanley J Assessment of potential bias in research grant peer review in Canada CMAJ 2018 April 23;190: 489-99)3. While the authors note that differences in scientific productivity may be the reason for differences in rating for male and female applicants, they have not measured it. This is a major limitation of this study that needs to be identified in the discussion4. There was a significant gender institutional affiliation interaction which was not discussed which was far more significant than the gender*gender interaction5. The methods section needs to be re-organized. The data and variables section should include the definitions of research topic, institutional affiliation (currently found in the statistical analysis section or supplements).6. The method used to calculate the application score needs to be defined. Are the criteria weighted equally? How are scores from the reviewers combined?7. Pg 10, only interactions from the multivariate analysis should be reported as there is confounding by research area and affiliation8. Pg 11, the discussion about why there are fewer women in science particularly some areas of science is irrelevant to the topic of this manuscript9. Table 1. The column percent should be presented to facilitate comparison rather than the row percent.
---

	10. Figure 1 Predicted values from the multivariate model should be presented as this figure over-inflates the difference in rating by gender by not adjusting for confounders
--	--

REVIEWER	Taeko Becque University of Southampton, UK
REVIEW RETURNED	03-Feb-2020

GENERAL COMMENTS	A comprehensive and well reported analysis of peer review scores of grant applications to the SNSF. Please clarify the main model that was fitted using an equation in the main text. (It is clear from the R code in the analysis notebook, which was helpful.) Given the very large sample size, it is not too surprising that small p values are obtained, but please comment on the magnitude of the effect sizes, e.g., an adjusted difference of 0.08 for male vs female applicants is very small compared to the standard deviation. Please report the effect sizes for the adjusted interaction models in Table S1, and comment that the gender interaction term, while significant, is also very small. Is it possible to present the graphs on a common scale, and "zoom out" a bit? (Ideally graphs would be on the original scale of 1 - 6, but in this case they could become illegible) In the discussion, please elaborate a little on why Marsh et al obtain no gender difference with a different statistical model.
---

VERSION 1 – AUTHOR RESPONSE

Reviewer: 1

Reviewer Name: Dr Robyn Tamblyn

Institution and Country: McGill University, CANADA

Please state any competing interests or state 'None declared': None declared.

This manuscript provides interesting policy relevant results about potential biases in peer review of grants.

Thank you.

Recommended modifications

1. There are sentinel articles on gender effects of reviewers and applicants that are missing from the reviewed literature.

Thank you. We now cite additional, more recent studies, including studies of grant applications submitted to the Canadian Institutes of Health Research, the National Institutes of Health and the European Research Council. Also, we re-organized the Introduction section and moved the discussion of the role of productivity to the Discussion section.

2. Both Wenneras and Tamblyn used scientific productivity measures to adjust for differences in the quality of the applicants. (Wennerås C, Wold A. Nepotism and sexism in peer-review. Nature 1997;387:341-3 , Tamblyn R, Girard N, Qian C, Hanley J Assessment of potential bias in research grant peer review in Canada CMAJ 2018 April 23;190: 489-99)

Both these papers are now cited in the Discussion.

3. While the authors note that differences in scientific productivity may be the reason for differences in rating for male and female applicants, they have not measured it. This is a major limitation of this study that needs to be identified in the discussion

We agree. We have identified this limitation in the Discussion section of our paper.

4. There was a significant gender institutional affiliation interaction which was not discussed which was far more significant than the gender*gender interaction

We added a paragraph on this interaction effect in the Discussion.

5. The methods section needs to be re-organized. The data and variables section should include the definitions of research topic, institutional affiliation (currently found in the statistical analysis section or supplements).

We have re-organized the Methods section as suggested.

6. The method used to calculate the application score needs to be defined. Are the criteria weighted equally? How are scores from the reviewers combined?

We clarified this in the revised Methods section. The external reviewers grade each of the criteria separately and then give an overall score themselves. The SNSF does not give any recommendations or guidelines on how the criteria should be weighted or summarized in the overall grade. The (internal) referee and co-referee can base their decision on all the grades they got from the external reviewers.

7. Pg 10, only interactions from the multivariate analysis should be reported as there is confounding by research area and affiliation

We now report the details on both analyses in Figure 1. Please note that the STROBE reporting guidelines for observational studies states that “unadjusted estimates and, if applicable, confounder-adjusted estimates” should be given.

8. Pg 11, the discussion about why there are fewer women in science particularly some areas of science is irrelevant to the topic of this manuscript

We have removed this part from the discussion.

9. Table 1. The column percent should be presented to facilitate comparison rather than the row percent.

We changed the percentages to column percentages.

10. Figure 1 Predicted values from the multivariate model should be presented as this figure over-inflates the difference in rating by gender by not adjusting for confounders

The predicted values from the multivariable model are now also shown.

Reviewer: 2

Reviewer Name: Taeko Becque

Institution and Country: University of Southampton, UK

Please state any competing interests or state 'None declared': None declared

A comprehensive and well reported analysis of peer review scores of grant applications to the SNSF.

Thank you.

Please clarify the main model that was fitted using an equation in the main text. (It is clear from the R code in the analysis notebook, which was helpful.)

A general equation of the mixed effects models fitted throughout the analysis is now included in the section of the Methods on the statistical analysis.

Given the very large sample size, it is not too surprising that small p values are obtained, but please comment on the magnitude of the effect sizes, e.g., an adjusted difference of 0.08 for male vs female applicants is very small compared to the standard deviation.

Indeed, the large sample size made it possible to detect small effects. The adjusted difference is small on average but indicates that there may be gender bias, which could have disadvantaged some women. In the revised version of the manuscript we stress that the adjusted differences is small but also make the point that the confounding by disciplines and institutions is a consequence of the "leaky pipeline", where women drop out of academia at a faster rate than men.

Please report the effect sizes for the adjusted interaction models in Table S1, and comment that the gender interaction term, while significant, is also very small.

Instead of adding the interaction effect sizes in the table, we decided to show the predicted values of the overall score from both the bivariable and multivariable models in Figure 1. The effect sizes of the interaction terms are reported in the online notebook.

Is it possible to present the graphs on a common scale, and "zoom out" a bit? (Ideally graphs would be on the original scale of 1 - 6, but in this case they could become illegible)

Thank you. We understand your concern and tried to make it clear in the figure legend that the scale goes from 1-6. We 'zoomed out' a bit, also to harmonize the axes of the bivariable analysis with the (newly added) multivariable analysis.

In the discussion, please elaborate a little on why Marsh et al obtain no gender difference with a different statistical model.

The discrepant results of the original meta-analysis by Bornmann et al (2007) and the re-analysis of the same data by Marsh et al (2009) were due to the different statistical models used. However, the multi-level approach employed by Marsh et al (2009) is not widely used and most of the individual studies showed a lower success rate in women compared to men, although effects tended to be small and failed to reach conventional levels of statistical significance. We think this is a minor point and not central to our study. In the revised version of our paper, we focus on the more recent literature and have removed the reference to Marsh et al's re-analysis. See also comments by reviewer 1.

VERSION 2 – REVIEW

REVIEWER	Taeko Becque University of Southampton, UK
REVIEW RETURNED	25-Mar-2020
GENERAL COMMENTS	All points have been addressed.